# Tracking single particles for hours via continuous DNA-mediated fluorophore exchange

Florian Stehr 🔘 [1,4], Johannes Stein 🔘 [1,4], Julian Bauer 🔘 [1], Christian Niederauer 🔘 [2], Ralf Jungmann 🔘 [1,3], Kristina Ganzinger 🔘 [2✉] & Petra Schwille 🔘 [1✉]

Monitoring biomolecules in single-particle tracking experiments is typically achieved by employing fixed organic dyes or fluorescent fusion proteins linked to a target of interest. However, photobleaching typically limits observation times to merely a few seconds, restricting downstream statistical analysis and observation of rare biological events. Here, we overcome this inherent limitation *via* continuous fluorophore exchange using DNA-PAINT, where fluorescently-labeled oligonucleotides reversibly bind to a single-stranded DNA handle attached to the target molecule. Such versatile and facile labeling allows uninterrupted monitoring of single molecules for extended durations. We demonstrate the power of our approach by observing DNA origami on membranes for tens of minutes, providing perspectives for investigating cellular processes on physiologically relevant timescales.

[1] Max Planck Institute of Biochemistry, Martinsried, Germany. [2] AMOLF, Amsterdam, The Netherlands. [3] Faculty of Physics, Ludwig Maximilian University, Munich, Germany. [4] These authors contributed equally: Florian Stehr, Johannes Stein. ✉email: k.ganzinger@amolf.nl; schwille@biochem.mpg.de

Single-Particle Tracking (SPT) is a powerful technique to study the motion and interactions of biomolecules in cellular or biomimetic environments[1,2]. Following dynamics and orchestration of molecular processes one molecule at a time has been key for developing the mechanistic concept of proteins as molecular machines[3]. Already since the use of colloidal gold particles as SPT labels[4,5], it became clear that both label size, photostability and target attachment are crucial factors to extract biologically meaningful data. To date, SPT mainly employs single fluorophores or quantum dots (QDs) as labels. More recently, also advanced multi-fluorophore labeling implementations have been developed[6,7]. Organic dyes are small and straightforward to attach to a target, but as their photon budget is limited, observations are only possible for a few seconds at 20–50 nm spatial precision before they photobleach[1]. Oxygen scavenging systems can improve the photon yield of organic dyes[8–12], but they are mostly incompatible with live-cell experiments[13]. QDs can overcome this limitation of fluorophores, as they are brighter and resistant to photobleaching[14]. However, QDs blink, and biocompatible QDs are large (~10–40 nm), potentially impairing the dynamics of the labeled molecules[1,14]. Furthermore, they are difficult to functionalize at the desired 1:1 stoichiometry[1], possibly resulting in artificial cross-linking of multiple molecules. Observing particle trajectories for an extended amount of time with high spatiotemporal resolution is however key to further our ability to extract physiologically meaningful data to observe rare biological events and improve theoretical models in the future[15].

Here, we introduce a labeling approach by re-purposing DNA-PAINT[16] (Points Accumulation for Imaging in Nanoscale Topography) to generate fluorescent labels with increased lifetimes, while maintaining live-cell compatibility, 1:1 labeling, and smaller sizes than QDs (see Supplementary Fig. 1 for size comparison). The key principle is DNA-mediated fluorophore exchange: short fluorescently-labeled oligonucleotides ('imagers', 8 bp; sequence: 5′-GAGGAGGA-3′-Cy3B) transiently bind a complementary single, long DNA strand attached to the molecule of interest ('tracking handle' TH, 54 base pairs, Fig. 1a). Imagers bind *via* reversible DNA hybridization of their short complementary oligonucleotide part, and both TH's 3′ and 5′ ends can be modified with functional groups for target labeling (e.g., thiol or click-chemistry, SNAP/HALO-tag, etc.). TH and imager sequences are designed such that one imager is bound to the TH at all times, with turnover being sufficiently fast to replace imagers before photobleaching occurs. This is achieved by allowing multiple (max. 6, see Supplementary Fig. 2) imagers to bind simultaneously, maximizing their association rate ($k_{on}$)[17–21], and optimizing their dwell times ($k_{off}$) (see Fig. 1b for schematic depiction).

## Results

### A continuously rejuvenating fluorescent label

To demonstrate the improved properties of the TH, we compared it to single Cy3B molecules fixed to DNA origami ('single-dye origami' or SD origami). We acquired images of immobilized target molecules at low surface densities via TIRFM[22] (Total Internal Reflection Fluorescence Microscopy; see Supplementary Table 1 for imaging conditions of all presented data). Both SD and TH origami data sets were then subjected to the same post-processing procedure (Supplementary Fig. 3 & 4). While SD origami bleached on the time scale of tens of seconds—after 2 min, nearly all dyes had entered a permanent dark state—a large fraction of TH origami was still observable even after 30 min, using identical acquisition conditions (Fig. 1c, d).

For quantitative comparison of single dyes and THs, we distinguish between the full trajectory (x, y, t) of a molecule and the sections where this molecular path can be visualized by detecting a fluorescence signal (see Fig. 1e), leading to the recording of (potentially) a multitude of trajectories per particle. Irrespective of the position coordinates (x, y), we hence introduce a metric that reflects not only on the measured trajectory durations ($\tau$) but also on the number of recorded trajectories per particle (TPP). We hence calculate $TPP_i(\tau \geq T)$, i.e., the number of trajectories with durations $\tau$ longer or equal to arbitrary query times $T$ for each immobilized DNA origami $i$ ($i = 1, 2, ..., M$, where $M$ denotes the total number of molecules after filtering). In contrast to permanent labeling with a single dye, where TPP is ideally expected to be 1, for the TH origami the TPP will grow with the duration of the measurement, as imagers are continuously replenished and a single TH can be repeatedly observed. Thus, a typical SD origami fluoresces from the start of image acquisition until it abruptly photobleaches (Fig. 1f), while a typical fluorescence trace of a TH origami shows an almost continuous signal of fluctuating intensities with short interruptions of a few frames (when no emitting imager is bound) resulting in five trajectories of durations $\tau_{1-5}$ in the range of ~10–200 s in the example shown (Fig. 1g; a 10 min subset of the trace is displayed for illustration purposes). Calculation of $TPP_i(\tau \geq T)$ for all $M$ fluorescence traces of the data set and subsequent averaging yielded the ensemble mean $TPP(\tau \geq T) = \frac{1}{M}\left(\sum_{i=1}^{M} TPP_i(\tau \geq T)\right)$ (Fig. 1h,i black line). We refer to the time at which the ensemble mean falls below one-half (i.e., $TPP\left(\tau \geq T_{1/2}\right) = 0.5$) as the characteristic half-life time $T_{1/2}$. Hence, the mean $TPP(\tau \geq T)$ and its corresponding $T_{1/2}$ simultaneously allow both a quantitative description of the number of trajectories obtained per particle/origami and their expected average duration. In other words, all ~3000 SD origami in the data set produced on average a single trajectory (Fig. 1h, $y$-axis intercept at 1) and half of these (i.e., 1500) had a duration of at least 11 s ($T_{1/2} = 11$ s). In contrast, each of the ~2500 TH origami yielded ~22 trajectories on average over a measurement duration of 10 min and $T_{1/2}$ analysis revealed that we registered 1250 trajectories with a duration of at least 200 s (>3 min), resulting in an increase in both the number of tracks and in $T_{1/2}$ of a factor of ~20× compared to SD origami (Fig. 1i). When the number of trajectories per frame was normalized to the first frame, we found that more than 80% of all THs were still detectable (fluorescent) when imaging for 30 min, while this was true for only ~3% of the SD origami already after 120 s (Fig. 1j). The 20% decrease in TH detection over time is likely due to photo-induced damage to the DNA caused by reactive oxygen species[21,23] during imaging (see Supplementary Note 1). To assess the number of imagers simultaneously bound to the TH, we analyzed the photon count distribution, which yielded distinct, equidistant single dye steps with the first step exhibiting the same photon count value in the TH and SD origami data sets (Supplementary Fig. 5). The increase in photons per localization in the TH case also resulted in a twofold increase in localization precision compared to SD origami (5.6 nm and 9.0 nm, respectively, Supplementary Fig. 5). In summary, compared to a single dye, a single TH produces: (1) more trajectories, which on average have (2) a longer duration and (3) a higher fluorescence brightness. Using a buffer compatible with live-cell imaging and optimized imaging conditions (T = 21° and 40 nM imager concentration), we obtained a 26-fold increase in $T_{1/2}$ for TH compared to SD origami (Fig. 1k, see Supplementary Note 1 for a

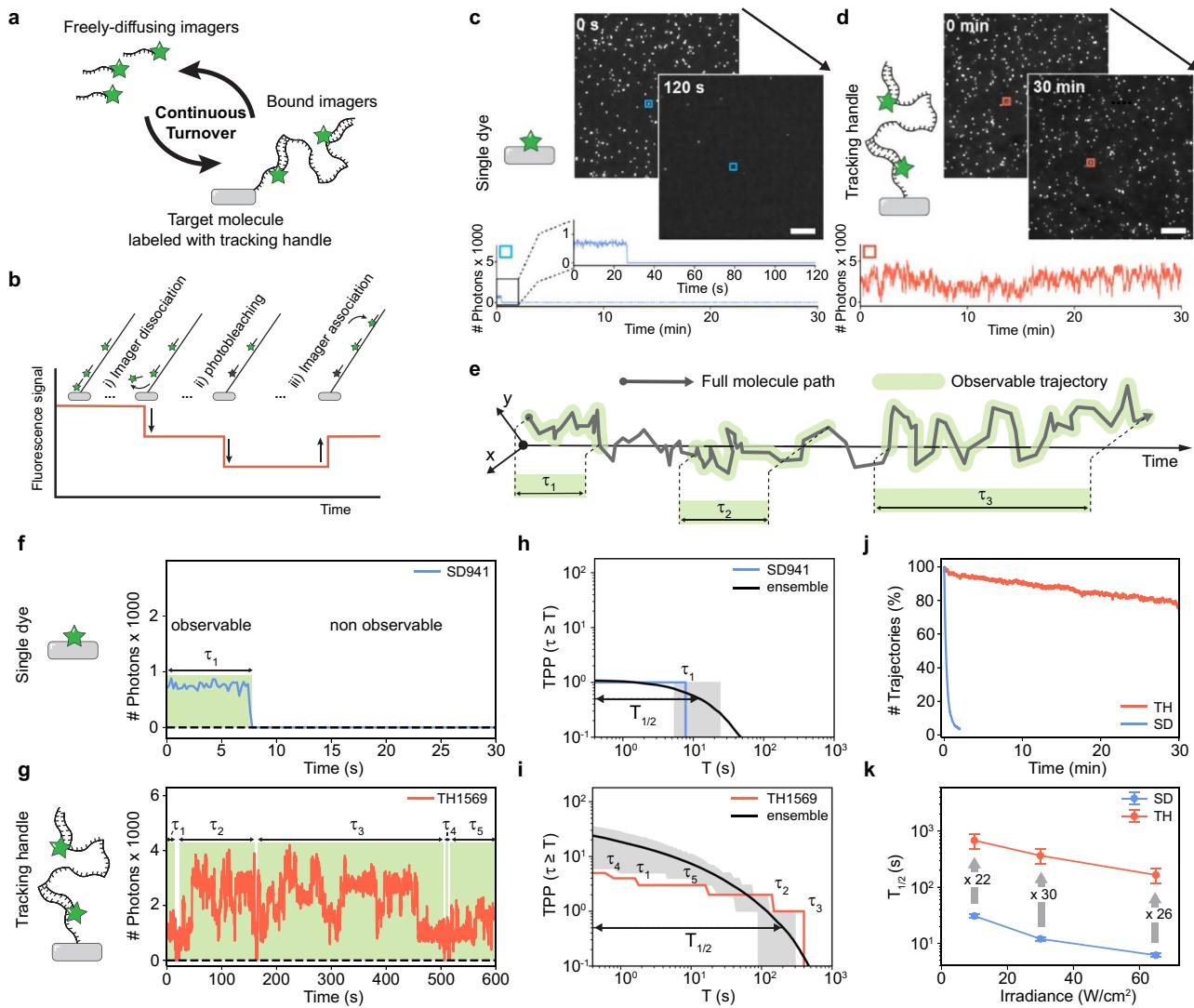

**Fig. 1 DNA-mediated continuous fluorophore exchange leads to long observations times for labeled molecules. a** Principle of re-purposing DNA-PAINT for single-particle tracking experiments. Freely-diffusion imagers are binding and unbinding in a continuous turnover to the TH attached to the target molecule. **b** Schematic of intensity fluctuations recorded from a TH caused by imager dissociation, association and photobleaching. **c** TIRFM imaging of static DNA origami labeled with single-dye (left) and fluorescence traces corresponding to the marked particle. **d** Same as (**c**) but for TH origami under equal imaging conditions and 40 nM of imager in solution. **e** Distinction between full molecular path (x-y-t) and its observable sections by means of a detectable fluorescence signal. This leads to the recording of a multitude of trajectories per particle of durations $\tau_i$. **f** SD example fluorescence trace (SD941 arbitrarily selected out of all ∼ 3000 SD origami surpassing the filter criteria described in Supplementary Fig. 4). **g** Arbitrarily selected TH example fluorescence trace (analogous to **f**). **h** Plot of $TPP(\tau \geq T)$ vs. T for the SD fluorescence trace in (**f**) (blue) and averaged over all SD origami in the data set (black). **i** Plot of $TPP(\tau \geq T)$ vs. T for the TH fluorescence trace in (**g**) (orange) and averaged over all TH origami in the data set (black). **j** Number of trajectories per frame (i.e., emitting labels) vs. measurement time normalized to initial trajectory number. Total number of recorded trajectories: n ∼149,000 (TH) and n ∼3700 (SD) (**k**) $T_{1/2}$ vs. irradiance plots for SD origami (blue) and TH origami (orange) imaged at varying irradiances without using an oxygen scavenging system (after filtering, SD data sets contained at least n ∼3000 origami and TH data sets at least n ∼700 origami). Arrows indicates factors of increase of TH vs. SD. Scale bars, 5 $\mu$m in (**c**, **d**). Error in (**h**, **i**) refers to interquartile range indicated as gray shaded area. Error bars in (**k**) correspond to relative standard deviation (see Supplementary Fig. 6).

detailed description and analysis of the optimization process). For instance, for an irradiance of 30 W/cm$^2$, we obtained $T_{1/2}$ of 365 s (>6 min) for TH compared to only 12 s for SD origami in live-cell-compatible conditions.

**Single particle tracking on supported lipid bilayers**. Next, we investigated the improvement of TH labeling for SPT of moving origami. We used eight biotin anchors to attach the origami *via* streptavidin to biotinylated lipids in supported lipid bilayers (SLB), mimicking two-dimensional diffusion (Supplementary

Movies 1 & 2; see Supplementary Fig. 7 for the analysis workflow). Comparing the 16 longest trajectories for SD and TH under identical conditions, we could readily observe that even the shortest trajectory measured for a TH origami (244 s) was two-times longer than any of the SD origami trajectories (<120 s, Fig. 2a, Supplementary Fig. 8 for different irradiances). Remarkably, we were able to track single TH origami for more than 18 min during a 30 min measurement. Similar to immobilized origami (see Fig. 1i), diffusing TH origami still yielded >50% of the initial trajectory number at the end of the measurement duration of 30 min (half time decay ∼ 33 min, Fig. 2b,

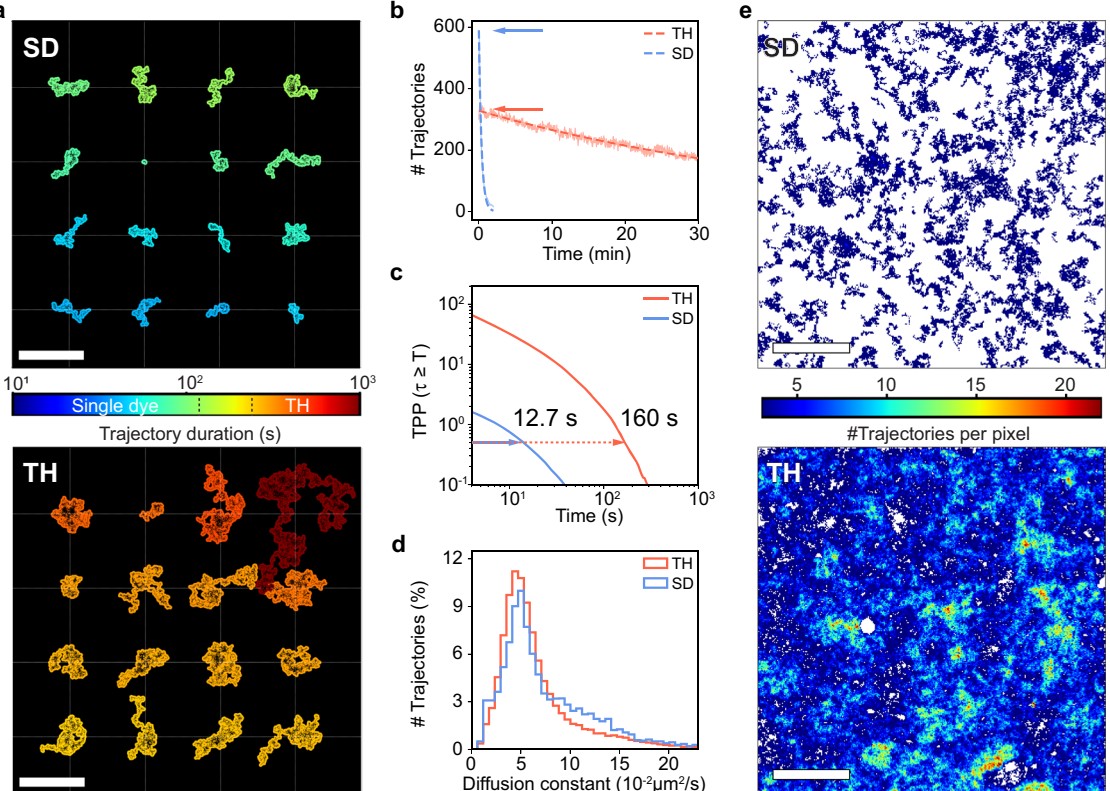

**Fig. 2 Probing 2D diffusion of DNA origami on lipid membranes. a** The 16 trajectories of longest duration of both SD origami (top) and TH origami (bottom) floating on an SLB. The trajectories where shifted according to their mean position indicated by the joints of the white dashed lines and were color-coded by duration for better visibility. **b** Number of trajectories per frame for SD origami (solid blue) and TH origami (solid orange). The curves were fitted by an exponential decay function (dashed). The arrows indicate the initial track number ($\sim$ number of Origami) $M$ within the FOV. **c** Average number of tracks per Origami $TPP(\tau_n \geq T)$ as obtained by normalization to the initial track number $M$ (arrows in **b**, see Supplementary Note 2). **d** Diffusion constants as obtained by linear iterative fitting of the individual MSD curves. Histograms represent the total distribution of three different samples imaged under three different irradiances. Only trajectories with more than 20 localizations were included giving a total of $n \sim$61.000 trajectories for the TH origami and $n \sim$9.000 trajectories for the SD origami. **e** Number of unique trajectories per 2 × 2 binned pixel (complete FOV). For each binned pixel we counted the number of unique trajectories passing through it during the entire measurement (i.e., a unique trajectory having at least one localization within the pixel boundaries increases its count by one). The TH origami allowed an almost complete mapping of the SLB (bottom) in contrast to only sparse sampling for the SD origami (top). White areas indicate that no trajectory passed these pixels over the complete measurement time. SLB tracking experiments were repeated at least three times for TH origami and at least nine times for SD origami yielding similar results (compare also varying irradiances in Supplementary Figs. 8 and 9). Scale bars, 10 $\mu$m in (**a**), 20 $\mu$m in (**e**).

Supplementary Fig. 9 for different irradiances). In total, we collected $\sim$19,000 trajectories for TH origami compared to only $\sim$900 for SD origami, representing a 20-fold increase in statistical sampling. For mobile SD origami we obtained a $T_{1/2}$ value of $\sim$12.7 s (Fig. 2c; see Supplementary Note 2 for mobile $TPP$ calculation) which is in good agreement with the $T_{1/2}$ of $\sim$11 s for immobilized SD origami (Fig. 1g; see Supplementary Fig. 9 for different irradiances). In contrast, we measured a $T_{1/2}$ of $\sim$160 s for TH origami, translating into more than 12-fold longer trajectories on average (Fig. 2c). The $\sim$2-fold reduction in TH trajectory lengths (observation times) compared to immobilized samples imaged under identical conditions is likely due to imperfect trajectory linking at the given particle densities (Supplementary Fig. 10). Even moving particles allowed to observe up to six discrete photon levels—corresponding to the number of imagers bound to the TH—which could potentially be used to resolve ambiguities during trajectory linking (Supplementary Fig. 11). The diffusion constant histograms of both SD and TH origami diffusing on SLBs, as obtained by linear (iterative) fitting of the individual mean square displacement (MSD) curves, agree well with each other, indicating that diffusion properties of origami are not altered by the TH (Fig. 2d). The relatively broad

range of diffusion constants ranging from 0.02 to 0.06 $\mu$m$^2$/s is likely caused by varying numbers of biotinylated anchors per TH origami due to a limited incorporation efficiency[24]. Finally, when we created spatially-resolved maps of single-molecule motions[25,26] of the entire FOV, both the high number of trajectories obtained per TH origami and the long duration of the individual trajectories allowed an almost complete mapping of SLB morphology with only $\sim$ 350 origami present in the FOV at the start of the measurement, in stark contrast to SD origami even when present at much higher initial particle densities (Fig. 2e; arrows in Fig. 2b).

**Long particle trajectories enable advanced quantitative analyses.** Apart from allowing us to efficiently map diffusional patterns in space, the significantly longer particle trajectories for TH origami also allowed us to analyze the motion of sub-trajectories to capture changes within a single trajectory. While promising important biological insights, such an analysis is statistically impossible for trajectories obtained from single-dye-labelled molecules. We divided all trajectories exceeding 120 s (600 frames) in duration into sub-trajectories of 10 s (50 frames), and applied the same MSD fitting algorithm to both the complete

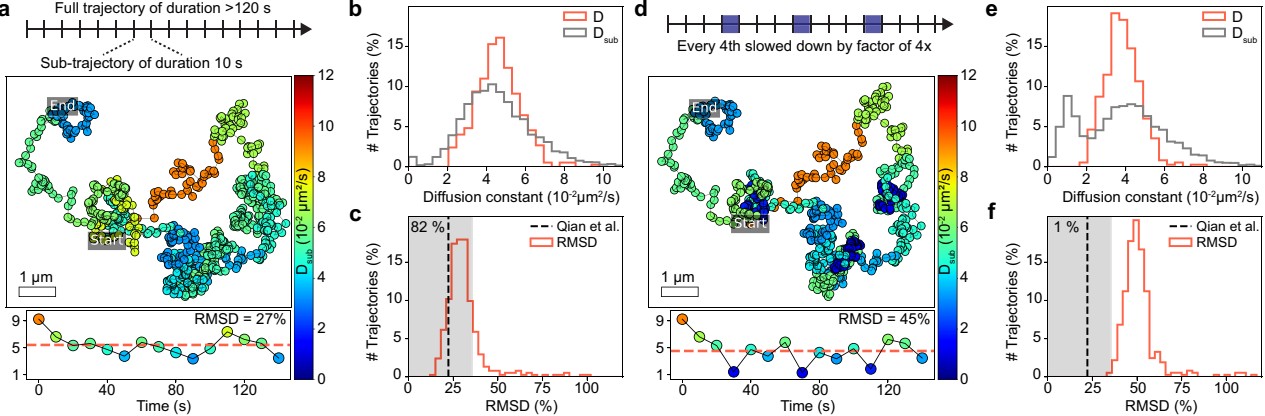

**Fig. 3 Subtrajectory analysis of TH origami diffusion on SLB.** Trajectories exceeding 120 s in duration were divided into sub-trajectories of duration 10 s and analyzed by the same MSD fitting procedure. **a** In the exemplary trajectory each localization is color-coded according to the obtained diffusion constant $D_{sub}$ of the corresponding subtrajectory. Time dependent scatter of $D_{sub}$ (see colorbar in **a**) around the diffusion constant $D$ (red dashed) as obtained from analysis of the full trajectory. **b** Total distribution of $D$ and the corresponding subtrajectory diffusion constants $D_{sub}$. **c** Root-mean-square deviation (RMSD) distribution of $D_{sub}$ to $D$ of all trajectories exceeding 120 s (red). RMSD was normalized to $D$ and should hence be close to the theoretical limit for the relative standard deviation of $D_{sub}$ (black dashed) if the TH origami are subject to a time-invariant Brownian motion (Supplementary Note 3). Gray area indicates deviation of <60% to the theoretical limit. (**d, e, f**) same as (**a, b, c**) but with sub-trajectories computationally slowed down as indicated in (**d**).

trajectory ($D$) and all of its sub-trajectories ($D_{sub}$, Fig. 3a). The diffusion coefficient distribution for both complete and sub-trajectories overlap perfectly indicating Brownian motion. The $D_{sub}$ distribution was 1.5-fold broadened due to the larger statistical uncertainty with decreasing trajectory lengths[27–29] (Fig. 3b, Supplementary Fig. 12). Indeed, we find that the scatter of $D_{sub}$ agrees very well with the statistically expected uncertainty for the given subtrajectory durations[27], confirming a time-invariant Brownian motion of the origami (Fig. 3c, Supplementary Fig. 13, see Supplementary Note 3 for details on the calculations). While time-invariant Brownian motion is expected for our experimental system, we tested if the applied analysis would indeed be capable of detecting motion changes between sub-trajectories, by computationally slowing down individual sub-trajectories (Fig. 3d). Such mobility changes are frequently observed in cell membranes and usually point to physiologically relevant local events[30]. A fourfold speed decrease in every fourth subtrajectory resulted in a second peak in the $D_{sub}$ distribution for the slower motion mode (Fig. 3e), and the scatter of $D_{sub}$ with respect to $D$ now clearly deviated from the statistically allowed limits for time-invariant Brownian motion (Fig. 3f, Supplementary Fig. 14).

**A practical handout—limitations and considerations.** In this study, we focused on reconstituted 2D systems that (1) featured a very thin excitation volume (TIRF) and (2) showed very low levels of unspecific binding[21]. This enabled us to use rather high imager concentrations of up to 40 nM. When shifting to cellular targets, one will most likely find a situation that deviates from these ideal conditions more or less strongly. In fact, fluorescence background from unbound diffusing imagers in solution and unspecific binding will dictate the upper bound of the imager concentration at which the TH can be operated in any biological system of interest. However, reducing the imager concentration also comes at the cost of shorter observation times of the TH. We characterized the effects on both the TH key observables and background fluorescence levels caused by either altering the excitation volume via variation of the TIRF angle (Supplementary Fig. 21) or by reducing imager concentrations (Supplementary Fig. 22).

One approach to regain longer trajectory durations even at lower imager concentrations is to label each target molecule with multiple THs. We therefore designed DNA origami with two TH extensions ($2 \times TH$) and compared the effect on the recorded trajectory durations to standard TH origami ($1 \times TH$) for varying imager concentrations (Supplementary Fig. 23). While the weight imposed on the target molecule by the $2 \times TH$ labeling is only doubled, the observation times are dramatically increased, allowing an eightfold reduction in imager concentration to measure similar observation times as for the $1 \times TH$ labeling. We reason that we can achieve a similar effect by extending the $1 \times TH$ handle sequence by multiples of the triplet CTC (e.g., from $18 \times$ to $36 \times$). Naturally, both ways lead to an increased size and weight of the label, potentially interfering with the dynamics of the target molecules at a certain point or reducing the achievable localization precision. However, we think that this approach can be a viable starting point for further optimization, especially in cases where background fluorescence and/or unspecific binding are the limiting factors.

In applications where live-cell compatibility is not required, usage of oxygen scavenging systems and triplet state quenchers strongly boost the performance of single dye molecules (Supplementary Fig. 15). In combination with increased imager dwell times at the TH (e.g., by increasing the imager length to $3 \times GAG$) this is an additional option to reduce the required imager concentrations.

## Discussion

In summary, our labeling strategy for fluorescence-based SPT, using a 1:1 functionalization with a DNA-based TH and exploiting DNA-mediated fluorophore exchange largely decouples trajectory lengths from the photon budget of single dye molecules, allowing for observations of target particles from minutes to hours depending on the experimental conditions. At the example of 2D diffusion on SLBs, we showed that the large number of trajectories nearly covered the FOV, which allowed mapping of the entire accessible membrane with an actual low number of particles. Even for moving THs, the number of currently bound imagers can be recovered from step-like intensity fluctuations in the fluorescence trace, which provides the

potential for intensity barcoding[31] and multiplexing[32] in the future beyond the use of orthogonal DNA sequences[33]. The ability to divide long trajectories into sub-trajectories paves the way for a robust quantitative analysis of the underlying motion dynamics[34–36], both in time and in space, and of molecular interactions, paramount to gaining mechanistic information into the biological systems studied by SPT.

We here demonstrated the strengths of our approach using an in vitro reconstituted system. However, we believe that the principle can be translated also to cellular targets, such as genetically-tagged[37,38] membrane proteins with accessible extracellular modification sites. Along this way, it will become particularly important to assess and minimize unspecific binding of negatively charged imagers with extracellular components. While intracellular targets of living cells are inaccessible to DNA-PAINT (due to degradation of imager strands by DNase), recently a peptide-based PAINT approach has been successfully demonstrated inside living cells[39,40], which could potentially enable genetic engineering of a peptide-based TH for intracellular targets.

A key challenge to overcome is the elevated background fluorescence currently limiting the tracking experiments to selective plane illumination schemes such as TIRF microscopy. Here, we have demonstrated that doubling the number of binding sites per particle (labeling with $2 \times$ TH) allowed a reduction in both imager concentration and background fluorescence, by eightfold and sixfold, respectively, without any loss in performance. Of course, this comes at the cost of additional load to the target molecules. A promising complementary solution to the problem of background fluorescence has recently been proposed in a study using self-quenching fluorogenic imagers for DNA-PAINT[41]. A combined implementation of fluorogenic imagers with our TH in a widefield configuration could allow, for instance, tracking of molecules also in 3D using point spread function engineering[42,43].

To conclude, our detailed analysis of the key parameters of DNA-PAINT based SPT will allow researchers to adapt the TH to their specific systems of interest, ranging from in vitro applications to potentially tracking receptors in the plasma membrane of living cells. We believe that due to its modularity and ease-of-use, our DNA-PAINT adaptation for SPT will become a valuable tool for studying dynamic processes at the single-molecule level.

## Methods

**Materials**. Unmodified, dye-labeled, and biotinylated DNA oligonucleotides were purchased from MWG Eurofins. DNA scaffold strands were purchased from Tilibit (cat. p7249, identical to M13mp18). Streptavidin was purchased from Thermo Fisher (cat. S-888). BSA-Biotin was obtained from Sigma-Aldrich (cat. A8549). All lipids were purchased from Avanti Polar Lipids. Glass slides were ordered from Thermo Fisher (cat. 10756991) and coverslips were purchased from Marienfeld (cat. 0107032). Freeze 'N Squeeze columns were ordered from Bio-Rad (cat. 732-6165). Tris 1M pH 8.0 (cat. AM9856), EDTA 0.5M pH 8.0 (cat. AM9261), Magnesium 1M (cat. AM9530G) and Sodium Chloride 5M (cat. AM9759) were ordered from Ambion. Ultrapure water (cat. 10977-035) was purchased from Thermo Fisher Scientific. Tween-20 (cat. P9416-50ML), Glycerol (cat. 65516-500ml), (+−)-6-Hydroxy-2,5,7,8-tetra-methylchromane-2-carboxylic acid (Trolox) (cat. 238813-5G), pyranose oxidase (PO, cat. P4234) and catalase (C, cat. C40) were purchased from Sigma-Aldrich. Two-component epoxy glue (cat. 886519 - 62) was purchased from Conrad Electronic SE.

**Buffers**. Seven buffers were used for sample preparation and imaging: Buffer A (10 mM Tris-HCl pH 7.5, 100 mM NaCl); Buffer B (5 mM Tris-HCl pH 8.0, 10 mM MgCl$_2$, 1 mM EDTA); Buffer L (20 mM HEPES pH 7.6, 140 mM NaCl, 3 mM MgCl2); Buffer POCT (5 mM Tris-HCl pH 8.0, 10 mM MgCl2, 1 mM EDTA, incubated 1 h prior to measurement with 1× PO, 1× C, 0.8% Glucose and 1× Trolox as previously described[23]); 10× folding buffer (100 mM Tris, 10 mM EDTA pH 8.0, 125 mM MgCl2); Buffer M (25 mM Tris-HCl pH 7.5, 150 mM KCl, 5 mM MgCl2); SLB buffer (25 mM Tris-HCl pH 7.5, 150 mM KCl).

**DNA origami design and assembly and purification**. DNA origami structures were designed using the design module of Picasso[16]. Our DNA origami design is identical to the one used in previous work[44], i.e., of flat rectangular geometry with just a single extension on the top side (at position 2B07 of Picasso Design). At this position, we as the label either incorporated the TH or a Cy3B molecule permanently attached to a short T-spacer, or a single DNA-PAINT docking strand (1DS) (see Supplementary Table 2 for sequences). On the bottom side, eight biotinylated extensions were incorporated for surface immobilization/SLB binding. Folding of structures was performed using the following components: single-stranded DNA scaffold (0.01 μM), core staples (0.1 μM), biotin staples (1 μM for SD origami and 0.01 μM for TH origami and 1DS origami), TH/SD/1DS strands (1 μM), 1× folding buffer in a total of 50 μl for each sample. Annealing was done by cooling the mixture from 80 to 25 °C in 3 h in a thermocycler. TH origami and 1DS were not purified after folding. SD origami were purified using PEG precipitation[45].

### Sample preparation

*Surface-immobilized DNA origami*. DNA origami samples were prepared as described before[16]. A glass slide was glued onto a coverslip with the help of double-sided tape (Scotch, cat. no. 665D) to form a flow chamber with inner volume of ~20 μl. First, 20 μl of biotin-labeled bovine albumin (1 mg/ml, dissolved in buffer A) was flushed into the chamber and incubated for 3 min. The chamber was then washed with 40 μl of buffer A. 20 μl of streptavidin (0.5 mg/ml, dissolved in buffer A) was then flushed through the chamber and incubated for 3 min. After washing with 40 μl of buffer A and subsequently with 40 μl of buffer B, 20 μl of biotin-labeled DNA origami (dilution from DNA origami stock dependent on origami yield after gel purification. Adjusted for each origami species individually to obtain a sparse DNA origami surface density. Starting dilution ~1:200) were flushed into the chamber and incubated for 10 min. The chamber was washed with 80 μl of imaging buffer (L/B/POCT) to remove unbound DNA origami. SD origami samples were sealed with two-component epoxy glue before imaging. For TH and 1DS origami samples, 40 μl of the imager solution was flushed into the chamber before sealing.

*DNA origami diffusing on supported lipid bilayers*. A glass slide was rubbed and rinsed with EtOH and ddH$_2$O and subsequently cleaned using a plasma cleaner (Zepto, Diener Electronic, Germany) for 40 s at 50% power and 0.3 mbar with oxygen as process gas. The glass slide was glued onto a coverslip with the help of double-sided tape (Scotch, cat. no. 665D) to form a flow chamber with inner volume of ~20 μl. Small unilamellar vesicles (SUVs) were prepared at a concentration of 4 mg/ml in buffer M from a lipid composition of 99 mol % DOPC/1 mol % Biotinyl-CAP-PE. Lipids dissolved in chloroform were dried under a stream of nitrogen. Vials were placed in a desiccator for 30 min to remove residual chloroform. After lipids were rehydrated in 200 μl of buffer M, the vials were placed in a sonicator bath to generate SUVs until the solution appeared transparent (storage of SUV solution aliquots possible at −30 °C for several weeks. After thawing of an aliquot it was placed in the sonication bath for 30 min). SUV solution was diluted to 0.5 mg/ml in buffer M. 20 μl of SUV dilution was flushed into the chamber and incubated for 3 min. The chamber was then washed with 5 × 80 μl of SLB buffer to remove excess vesicles and 1 × 80 μl with buffer B. 20 μl of streptavidin (0.5 mg/ml, dissolved in buffer A) was then flushed through the chamber and incubated for 5 min. After washing with 80 μl of buffer B, 20 μl of biotin-labeled DNA origami was flushed into the chamber and incubated for 3 min (origami dilution after folding ~1:1,000). Excess DNA origami were washed with 80 μl of buffer L. SD origami samples were sealed with two-component epoxy glue before imaging. For TH origami samples 40 μl of the imager solution were flushed into the chamber before sealing.

**Super-resolution microscopy setup**. Fluorescence imaging was carried out on an inverted custom-built microscope (see supplementary references[44,46] for detailed sketches) in an objective-type TIRF configuration with an oil-immersion objective (Olympus UAPON, 100×, NA 1.49). One laser was used for excitation: 561 nm (1 W, DPSS-system, MPB). The laser power was adjusted via polarization rotation using a half-wave plate (Thorlabs, WPH05M-561) before passing a polarizing beam-splitter cube (Thorlabs, PBS101). The laser light was coupled into a single-mode polarization-maintaining fiber (Thorlabs, P3-488PM-FC-2) using an aspheric lens (Thorlabs, C610TME-A) in order to spatially clean the beam-profile. Using a zero-order half-wave plate (Thorlabs, WPH05M-561) the coupling polarization into the fiber was adjusted. The laser light was re-collimated after the fiber using an achromatic doublet lens (Thorlabs, AC254-050-A-ML) resulting in a collimated FWHM beam diameter of ~6 mm. The Gaussian laser beam-profile was transformed into a collimated flat-top profile using a refractive beam shaping device (AdlOptica, piShaper 6_6_VIS). The laser beam diameter was magnified by a factor of 2.5 using a custom-built telescope (Thorlabs, AC254-030-A-ML and Thorlabs, AC508-075-A-ML). The laser light was coupled into the microscope objective using an achromatic doublet lens (Thorlabs, AC508-180-A-ML) and a dichroic beam splitter (AHF, F68-785). Fluorescence light was spectrally filtered with a laser notch filter (AHF, F40-072) and a bandpass filter (AHF Analysentechnik, 605/64) and imaged on a sCMOS camera (Andor, Zyla 4.2) using a tube lens without further magnification (Thorlabs, TTL180-A) resulting in an effective pixel size of 130 nm (after 2 × 2 binning). Microscopy samples were

mounted into a closed water-based temperature chamber (Okolab, H101-CRYO-BL) on an x-y-z stage (ASI, S31121010FT and ASI, FTP2050) that was used for focusing with the microscope objective being at fixed position. The temperature of the objective was actively controlled using the same water cycle as the temperature chamber. Focus stabilization was achieved via the CRISP autofocus system (ASI @ 850 nm) in a feedback loop with a piezo actuator (Piezoconcept, Z-INSERT100) moving the sample. The CRISP was coupled into the excitation path of the microscope using a long pass dichroic mirror (Thorlabs, DMLP650L). Our custom TIRF setup was used for all presented data.

**Imaging conditions.** All fluorescence microscopy data was recorded with our sCMOS camera (2048 × 2048 pixels, pixel size: 6.5 µm). The camera was operated with the open source acquisition software µManager[47] at 2 × 2 binning and cropped to the center 700 × 700 pixel FOV. The exposure time was set to 200 ms, the read out rate to 200 MHz and the dynamic range to 16 bit. The laser power was set to a homogeneous (flat-top profile, see setup description) with a measured excitation beam diameter of d ∼ 130 $\mu m$[46]. We performed all experiments at laser excitation powers $P$ of 1.4 mW, 4.1 mW and 8.9 mW (as measured at the fiber exit port). We hence calculated an (upper limit) homogeneous excitation irradiance $E$ at the sample space of 10 W/cm², 30 W/cm² and 65 W/cm² by using $E = 4P/\pi d^2$. SD origami samples were imaged repeatedly at 3× field of views (FOVs) for increased statistics, TH origami samples only at 1× FOV. For detailed imaging parameters specific to the data presented in all main and supplementary figures refer to Supplementary Table 1.

**Image processing & single particle tracking analysis.** Please refer to Supplementary Fig. 3, Supplementary Fig. 4 and Supplementary Fig. 7 for a step-by-step illustration through all processing steps of immobilized and diffusing DNA origami data, respectively. In any case, a standard single molecule localization algorithm was applied to the raw image stack to obtain a pointillist super-resolution image of the DNA origami[16] (picasso, see below). Immobile DNA origami appeared as localization clusters after image correlation based undrifting[16]. Subsequently, we automatically detected all localization clusters as previously described[44] and extracted the corresponding fluorescence traces (see e.g., Fig. 1f,g) for further analysis (spt, see below). Note that for all surface-immobilized experiments (for both SD origami and TH origami) we ignored interruptions in the fluorescence trace of just a single frame when determining the trajectory durations (i.e., 'ignore' in Supplementary Fig. 3). In the case of moving origami a nearest neighbor based linking algorithm (trackpy, see below) was applied to the localized raw image stack to obtain particle trajectories. Subsequent MSD analysis of the individual trajectories was carried out using custom python code (spt, see below).

The described analysis workflow was completely based on two custom written python packages https://github.com/schwille-paint/picasso_addon (picasso_addon[48]) and https://github.com/schwille-paint/SPT (spt). These packages integrate https://github.com/jungmannlab/picasso (picasso[49]) for localization of raw image stacks and on https://soft-matter.github.io/trackpy/v0.4.2/ (trackpy[50]) for linking of localizations into particle trajectories. The custom packages picasso_addon and spt hence provide a complete single particle tracking analysis suite for both mobile and immobilized particles. Please visit https://picasso-addon.readthedocs.io/en/latest/index.html and https://spt.readthedocs.io/en/latest/index.html for detailed information about the picasso_addon and spt API.

**Reporting summary.** Further information on research design is available in the Nature Research Reporting Summary linked to this article.

## Data availability
Source data are provided with this paper. The raw data that support the findings of this study are available from the corresponding authors upon reasonable request. Source data are provided with this paper.

## Code availability
Custom written python modules packages were employed in this study, which are available in public repositories: https://github.com/schwille-paint/picasso_addon[48] and https://github.com/schwille-paint/SPT[49].

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

## Acknowledgements

We thank Beatrice Ramm, Tamara Heermann, Henri Franquelim, Patrick Schueler, Sigrid Bauer and Katharina Nakel for experimental support and helpful discussions. J.S. and F.S. acknowledge support from Graduate School of Quantitative Bioscience Munich (QBM). All authors acknowledge support from the Center for Nano Science (CeNS).

This work has been supported in part by the German Research Foundation through the Emmy Noether Program (DFG JU 2957/1-1 to R.J.), the SFB1032 (projects A11 and A09 to R.J. and P.S.), the European Research Council through an ERC Starting Grant (MolMap; grant agreement number 680241 to R.J.) and the Max Planck Society (P.S. and R.J.).

## Author contributions

F.S., J.S., R.J., K.G. and P.S. conceived the study. F.S., J.S. and J.B. designed experiments. J.S. performed the experiments. J.B. devised the concept of the tracking handle sequence, prepared DNA origami and performed initial experiments. C.N. performed initial experiments. F.S. and J.S. designed and performed data analysis. F.S. wrote the analysis code. F.S., J.S., R.J., K.G. and P.S. wrote the paper. K.G. and P.S. supervised the study. All authors discussed and interpreted results and revised the paper.

## Funding

## Competing interests

The authors declare no competing interests.
