## [Peer Review File · Nature Communications]

Reviewers' Comments:

Reviewer #1:

Remarks to the Author:

The authors present an extension of DNA PAINT to single particle tracking. If there is one criticism, it would be that this is not a huge conceptual leap from the current state of the art - someone familiar with the field would reasonably expect it to work. This probably devalues the amount of effort required to actually get it working well and to characterise the technique, however, and the authors have been exceptionally thorough in this respect. The paper is a technical "tour-de-force" and I have no doubt that it will be a valuable resource for anyone wishing to perform similar DNA-PAINT tracking experiments.

I hesitate to suggest too many changes as I think the manuscript is in a good shape as it currently stands. I would however encourage the authors to address at least some of the following points. Many of these relate to being transparent about potential (minor) limitations of the technique - something I think should have higher prominence than is typically the case in published papers. I think that papers describing a new technique should have enough information about potential limitations so that a new graduate student who gets told to, e.g. work up the method in their PIs 200um thick Drosophila egg chambers, has some inkling of what they are up against.

Background

The PAINT based tracking data appears to have a higher background than conventional single fluorophore tracking (as would be expected from the diffusing unbound imager strands).

Does this effectively limit the technique to TIRF imaging?

How does the background effect 3D localisation precision (which is more sensitive to background than lateral localisation)?

What about imaging speed - is there an upper bound on imaging speed when individual diffusing imager strands start to get visualised and localised?

I suspect there is a somewhat complex trade-off between imager strand concentration, background, and site-occupancy / frequency of interrupted traces. If you have any data supporting your choice of imager strand concentration it would be great to add it to the supplement.

Cell compatibility

The authors only present results on reconstituted systems (lipid bilayers and targets immobilised on a coverslip).

Is there a realistic strategy/outlook for performing DNA-PAINT based tracking inside living cells? (I can see the potential for some fairly substantial technical issues in a) getting the target strand specifically attached to an intracellular target, b) getting the imager strands inside the cell, and c) protecting both from cellular DNases). In terms of practically addressing this in the manuscript I suspect there are a few reviews/perspectives on intracellular PAINT which could be cited.

When used to track cell surface targets (an easier proposition than intracellular targets), what is the chance that the significant -ve charge carried by the DNA will cause unwanted interactions with the extracellular matrix/glycocalyx? In reconstituted lipid bi-layers we would expect the negative membrane charge to help keep the label off the membrane and non-interacting, but this could change significantly on a proper cell membrane.

I should be clear that I think most of the above could be handled with a sentence or two in the discussion / supplement rather than extensive new experiments. I do think that tracking a cell surface protein and showing that it's diffusion was unchanged by the label would significantly strengthen the paper, but would likely recommend publication even in the absence of such data.

David Baddeley

Reviewer #2:

Remarks to the Author:

In "Tracking Single Particles for Hours via Continuous DNA-mediated Fluorophore Exchange" Stehr et al describe a nice way to extend tracking times in biomimetic systems, here supported lipid bilayers.

The approach is very interesting and could be very useful.

1) While it is suitable to characterise the approach in a fairly well-controlled environment it would be good to see it used in an actual cell system. Short of carrying this out as part of this work it would be useful for the authors to describe the most accessible/tractable cell system in which this could be explored as part of a brief discussion.

2) Please briefly mention the much increased tracking time with oxygen scavenging systems for the SD case in the main text. The point about lack of live cell compatibility is well taken but it would still be desirable to have this fact mentioned in the main text so that readers will notice without having to scrutinise the details of the supplementary info.

3) The "tracking handle" approach is clearly related to the use of repeated imager bindings sites that has been employed for imaging related purposes recently and variously termed 'sequence repeats' (Strauss & Jungmann, 2020), 'DNA-PAINT-ERS' (Civitci et al, 2020) or 'repeat DNA-PAINT' (Clowsley et al, 2021). These prior works mostly used repeated imager binding domains in the low imager concentration limit of \leq a single imager bound per strand. Clowsley et al, 2021, also explored the use of higher imager concentrations so that several imagers would be bound (used for diffraction-limited imaging), and also demonstrated the higher robustness against photo-induced docking site loss confirmed in this MS.

While the application in the current MS is novel, the relevant prior work should be appropriately described and cited, currently only Strauss & Jungmann seems covered.

4) Could even higher repeat numbers be beneficial for the tracking handle? A priori it not clear that 6 is "optimal" in any particular sense. Even though strands get longer the effective hydrated radius of the mostly single stranded handle grows slowly due to random coiling (as noted in Supplementary Figure 1) and higher repeat numbers could further increase robustness against photo-induced site loss. In connection with this point, more than one tracking handle could be employed per tracking particle. This would also allow lowering imager concentration further if desirable. Please discuss pros and cons.

5) In supplementary fig. 4 some of the description seemed difficult to follow. Particularly, the following sentences were hard to parse: "Second, all picks exceeding the 90% percentile of the number of events distribution of all picks are disregarded, i.e. all fluorescence traces consisting of more than 5 events in this case"; "For TH origami (illustration analogous to a) we divided the total number of localizations within a fluorescence trace by the measurement duration in frames

(occupation) that can be visualized as a compression of the overall event durations". Please consider reformulating. It seems odd to call this a compression.

REVIEWER COMMENTS

We would like to thank both reviewers for their thorough evaluations and for their positive feedback appreciating our technical efforts. Moreover, we are particularly grateful for the critical remarks regarding the limitations and potential applications. We think that these have helped us to significantly strengthen the manuscript, aiding future users in their own implementations of the Tracking Handle (TH). In the following, we address each comment individually and highlight the changes/additions in the manuscript.

Reviewer #1 (Remarks to the Author):

The authors present an extension of DNA PAINT to single particle tracking. If there is one criticism, it would be that this is not a huge conceptual leap from the current state of the art - someone familiar with the field would reasonably expect it to work. This probably devalues the amount of effort required to actually get it working well and to characterize the technique, however, and the authors have been exceptionally thorough in this respect. The paper is a technical “tour-de-force” and I have no doubt that it will be a valuable resource for anyone wishing to perform similar DNA-PAINT tracking experiments.

I hesitate to suggest too many changes as I think the manuscript is in a good shape as it currently stands. I would however encourage the authors to address at least some of the following points. Many of these relate to being transparent about potential (minor) limitations of the technique – something I think should have higher prominence than is typically the case in published papers. I think that papers describing a new technique should have enough information about potential limitations so that a new graduate student who gets told to, e.g. work up the method in their PIs 200um thick Drosophila egg chambers, has some inkling of what they are up against.

Background

1.1) The PAINT based tracking data appears to have a higher background than conventional single fluorophore tracking (as would be expected from the diffusing unbound imager strands). Does this effectively limit the technique to TIRF imaging?

As the reviewer correctly states, DNA-PAINT suffers from substantial background due to fluorescence from freely diffusing imager strands. For this reason, our approach requires some kind of selective plane illumination, at least in its current form. To clearly state this we added l. 144-145 to the results part:

“In this study, we focused on reconstituted 2D systems that 1) featured a very thin excitation volume (TIRF) and 2) showed very low levels of unspecific binding²¹.”

Additionally, we highlighted this point in l. 186-187 of the discussion:

“A key challenge to overcome is the elevated background fluorescence currently limiting the tracking experiments to selective plane illumination schemes such as TIRF microscopy.”

A possible solution to this problem might be the combination of the TH with recently proposed self-quenching fluorogenic imagers¹ as mentioned in l. 190-193 of the discussion:

“A promising complementary solution to the problem of background fluorescence has recently been proposed in a study using self-quenching fluorogenic imagers for DNA-PAINT⁴¹. A combined implementation of fluorogenic imagers with our TH in a widefield configuration could allow, for instance, tracking of molecules also in 3D using point spread function engineering^{42,43}.”

1.2) How does the background effect 3D localisation precision (which is more sensitive to background than lateral localisation)?

Most 3D localization approaches rely on the detection of subtle changes of the PSF (e.g., change in ellipticity of the PSF by insertion of a cylindrical lens). Even for homogeneous but elevated levels of background fluorescence, we expect that the detection of these changes will be less efficient. We hence agree with the reviewer’s assumption that 3D localization precision might be deteriorated in case of high imager concentrations as used in the current implementation of the technique. A full 3D implementation constitutes a major challenge at this point, due to the technique’s current limitation to selective plane illumination schemes. Therefore, we purely focused on the analysis of 2D systems in this work. To clearly state this we added l. 144-145 and l. 186-187 to the manuscript (see our answer to question 1.1).

Although we can so far not provide any 3D data to directly answer the reviewer’s question we added the dependency of measured signal & background photon levels for varying TIRF angles or varying imager concentrations in **Supplementary Fig. 21** and **Supplementary Fig. 22**. These can serve as a starting point for theoretical calculations or adaptations to the experimental system of choice (see l. 150-152):

*“We characterized the effects on both the TH key observables and background fluorescence levels caused by either altering the excitation volume via variation of the TIRF angle (**Supplementary Fig. 21**) or by reducing imager concentrations (**Supplementary Fig. 22**).”*

Supplementary Figure 21. Performance of tracking handle at varying TIRF angles. (a) Sketch to illustrate the variation of the TIRF angle λ defined as the angle of incidence of the excitation light with respect to the optical axis. In order to vary λ , a linear stage bearing the coupling lens focusing the laser into the back focal plane (BFP) of the objective was moved to laterally decrease the offset from the optical axis within the BFP. In our case, a positive displacement of the TIRF stage (bottom) with respect to the zero position (top) resulted in a decrease of λ . According to literature¹², a decrease in λ leads to an increasing penetration depth of the resulting evanescent field and hence a larger excitation volume (indicated by green arrows). Additionally, a decrease in λ results in higher excitation intensities (relative to the incoming light) of the resulting evanescent field (indicated by intensity vs. z plots to the right). (b) Plot of $TPP(\tau \geq T)$ vs. T for immobilized TH origami acquired at different stage displacements (i.e., varying λ). We observed shorter $T_{1/2}$ -values (given in brackets in the legend) with increased displacements (i.e., decreased λ) indicating a higher irradiance (compare **Supplementary Fig. 15** and see **Supplementary Table 1** for imaging conditions). (c) As explained in **Supplementary Fig. 4**, we calculated the occupancy for immobilized TH origami. The plot shows the complementary cumulative distribution function (CCDF=1-CDF) of the calculated occupancies for the three data sets in (a). As expected from (a), the occupancy decreases with increasing TIRF angle. (d) Top: Plot of mean registered background photons per localization vs. TIRF stage position for the three data sets in (b-c). A larger displacement (i.e., a smaller λ) results in higher background due to the increased excitation volume within the non-fluorogenic imaging solution. Middle: Plot of mean photons detected per localization vs. TIRF stage position also showing an increase with larger displacements (i.e., smaller λ), indicating an increased intensity of the evanescent (excitation) field. Bottom: Plot of signal-to-noise ratio (SNR) vs. TIRF stage position. Overall, the SNR (detected TH photons/background photons) decreases due to the more pronounced background with larger displacements (i.e., smaller λ). Error bars correspond to standard deviation.

Supplementary Figure 22. Performance of tracking handle at varying imager concentrations. (a) Plot of $TPP(\tau \geq T)$ vs. T for immobilized TH origami acquired at different imager concentrations. As expected we observed shorter $T_{1/2}$ -values (given in brackets in the legend) for decreasing imager concentration. (b) Complementary cumulative distribution function (CCDF) of occupancies for immobilized TH origami acquired at different imager concentrations (equivalent to **Supplementary Fig. 21, b**). (c) Top: Plot of mean registered background photons per localization vs. imager concentration for the data shown in (a-b) indicating linear response of the background fluorescence vs. imager concentration. The blue line indicates the measured value for SD origami for the same imaging conditions. Middle: Plot of mean photons detected per localization vs. imager concentration indicating saturation behavior for higher imager concentrations. Bottom: Plot of signal-to-noise ratio (SNR) vs. imager concentration. Overall, the SNR (detected TH photons/background photons) decreases due to the more pronounced background with higher imager concentrations. Error bars and dashed lines correspond to standard deviation.

1.3) What about imaging speed - is there an upper bound on imaging speed when individual diffusing imager strands start to get visualised and localised?

Recent publications reported free diffusion coefficients D in solution of at least $D = 150 \mu\text{m}^2/\text{s}$ for labeled DNA oligonucleotides of similar length as used in our experiments^{2,3}. We reason that individual diffusing imager strands could potentially be localized if their diffusion (area) A_D would not largely exceed the area covered by the PSF of the microscope A_{PSF} within the used exposure time. Hence, assuming a minimal exposure time of $t = 5 \text{ ms}$ for a camera-based system this would lead to an area of $A_D = Dt = 0.75 \mu\text{m}^2$ being explored by individual imager strands during this time. If we further assume a PSF with a FWHM $\approx 0.3 \mu\text{m}$ we can calculate an (upper bound) estimate of the PSF area of $A_{\text{PSF}} = \pi \times \text{FWHM}^2 = 0.28 \mu\text{m}^2$. Hence even using this simplistic model (that still neglects 3D diffusion and imager density/overlap) $A_D = 0.75 \mu\text{m}^2$ exceeds $A_{\text{PSF}} = 0.28 \mu\text{m}^2$ by almost a factor of three, which makes identification/localization of freely diffusing imager strands very unlikely. In conclusion, we reason that for realistic frame rates ($< 200 \text{ Hz}$) of camera based systems it will be very hard to approach the limit where individual unbound diffusing imagers start to get localized. Quite generally, our approach does not allow to study targets with diffusion coefficients approaching the diffusion coefficient of the imager strands, as the reviewer already anticipated. One interesting approach to overcome this upper bound of imaging speed that is compatible

with the TH is presented in the work of Chung et al.¹ and discussed in l. 190-193 of the discussion (see our answer to question 1.1)

1.4) I suspect there is a somewhat complex trade-off between imager strand concentration, background, and site-occupancy / frequency of interrupted traces. If you have any data supporting your choice of imager strand concentration it would be great to add it to the supplement.

We would like to thank the reviewer for this advice. In addition to **Supplementary Note 1** we have now added **Supplementary Fig. 22** and refer to it in l. 150-152 of the results (see answer to question 1.2):

Cell compatibility

The authors only present results on reconstituted systems (lipid bilayers and targets immobilized on a coverslip).

1.5) Is there a realistic strategy/outlook for performing DNA-PAINT based tracking inside living cells? (I can see the potential for some fairly substantial technical issues in a) getting the target strand specifically attached to an intracellular target, b) getting the imager strands inside the cell, and c) protecting both from cellular DNases). In terms of practically addressing this in the manuscript I suspect there are a few reviews/perspectives on intracellular PAINT which could be cited.

We thank the reviewer for raising these important points. We fully agree with the reviewer that all three potential issues mentioned currently prohibit to perform DNA-PAINT experiments inside living cells. To highlight this, we added l. 182-185 to the manuscript, further discussing possibilities toward a live-cell compatible adaptation of the TH on the basis of LIVE-PAINT⁴ or Peptide-PAINT⁵.

“While intracellular targets of living cells are inaccessible to DNA-PAINT (due to degradation of imager strands by DNase), recently a peptide-based PAINT approach has been successfully demonstrated inside living cells^{39,40}, which could potentially enable genetic engineering of a peptide-based TH for intracellular targets.”

We have cited the relevant publications to the best of our knowledge, but kindly ask the reviewer to point to any overlooked references discussing live cell DNA-PAINT limitations/implementations .

Due to the aforementioned issues, we believe that a more straightforward step toward a cellular application of our approach would ideally target a membrane bound molecule with accessible extracellular modification sites as mentioned in l. 179-181 of the discussion:

“However, we believe that the principle can be translated also to cellular targets, such as genetically-tagged^{37,38} membrane proteins with accessible extracellular modification sites.”

1.6) When used to track cell surface targets (an easier proposition than intracellular targets), what is the chance that the significant -ve charge carried by the DNA will cause unwanted interactions with the extracellular matrix/glycocalyx? In reconstituted lipid bi-layers we would expect the negative membrane

charge to help keep the label off the membrane and non-interacting, but this could change significantly on a proper cell membrane.

The sample surfaces in our experiments consisted of 1) oxygen-plasma cleaned glass passivated with biotinylated BSA or 2) almost pure (neutrally charged) DOPC lipid bilayers (99 mol %) on oxygen-plasma cleaned glass. Our literature screening suggests that neither of these surfaces carries any significant permanent negative charge^{6,7}. If the reviewer has any studies at hand pointing into a different direction, we would be very interested to know about in order to include them in the manuscript.

Despite this, we undoubtedly observed very low levels of unspecific binding in our experiments allowing us to use rather high imager concentrations. As the reviewer correctly states, one will most likely observe more unspecific binding in cellular environments. To discuss this and give some guidance in choosing an appropriate starting point for further optimization we added l. 147-152 to the results:

“In fact, fluorescence background from unbound diffusing imagers in solution and unspecific binding will dictate the upper bound of the imager concentration at which the TH can be operated in any biological system of interest. However, reducing the imager concentration also comes at the cost of shorter observation times of the TH. We characterized the effects on both the TH key observables and background fluorescence levels caused by either altering the excitation volume via variation of the TIRF angle (Supplementary Fig. 21) or by reducing imager concentrations (Supplementary Fig. 22).”

One additional step to further reduce imager concentrations without compromise in observation times was already proposed by Reviewer #2. Along this line, we now included newly acquired data presented in l. 153-162 of the results (see our response to question 2.4).

I should be clear that I think most of the above could be handled with a sentence or two in the discussion / supplement rather than extensive new experiments. I do think that tracking a cell surface protein and showing that it's diffusion was unchanged by the label would significantly strengthen the paper, but would likely recommend publication even in the absence of such data.

David Baddeley

Once again, we thank the reviewer for his positive evaluation and constructive feedback that initiated to seek for new answers to various critical aspects of this study.

Reviewer #2 (Remarks to the Author):

In “Tracking Single Particles for Hours via Continuous DNA-mediated Fluorophore Exchange” Stehr et al describe a nice way to extend tracking times in biomimetic systems, here supported lipid bilayers. The approach is very interesting and could be very useful.

We would like to thank this reviewer for the positive evaluation of our work.

2.1) While it is suitable to characterize the approach in a fairly well-controlled environment it would be good to see it used in an actual cell system. Short of carrying this out as part of this work it would be useful for the authors to describe the most accessible/tractable cell system in which this could be explored as part of a brief discussion.

Thank you very much for raising this important point, which is in line with Reviewer #1. We kindly refer Reviewer #2 to our answers to the questions 1.5 and 1.6 raised by Reviewer #1, where we have included a discussion regarding the most tractable cellular targets and potential difficulties to overcome along the way.

2.2) Please briefly mention the much increased tracking time with oxygen scavenging systems for the SD case in the main text. The point about lack of live cell compatibility is well taken but it would still be desirable to have this fact mentioned in the main text so that readers will notice without having to scrutinize the details of the supplementary info.

We agree with the reviewer that this useful information should be mentioned in the main text. We included the following sentence in the main text l. 163-166:

*“In applications where live cell compatibility is not required, usage of oxygen scavenging systems and triplet state quenchers strongly boost the performance of single dye molecules (**Supplementary Fig. 15**). In combination with increased imager dwell times at the TH (e.g., by increasing the imager length to 3×GAG) this is an additional option to reduce the required imager concentrations.”*

2.3) The “tracking handle” approach is clearly related to the use of repeated imager bindings sites that has been employed for imaging related purposes recently and variously termed 'sequence repeats' (Strauss & Jungmann, 2020), 'DNA-PAINT-ERS' (Civitci et al, 2020) or 'repeat DNA-PAINT' (Clowsley et al, 2021). These prior works mostly used repeated imager binding domains in the low imager concentration limit of \leq a single imager bound per strand. Clowsley et al, 2021, also explored the use of higher imager concentrations so that several imagers would be bound (used for diffraction-limited imaging), and also demonstrated the higher robustness against photo-induced docking site loss confirmed in this MS. While the application in the current MS is novel, the relevant prior work should be appropriately described and cited, currently only Strauss & Jungmann seems covered.

We thank the reviewer for pointing out that the references to the works of Civitci⁸ et al. and Clowsley⁹ et al., who already used the same rationale of docking strand design to maximize k_{on} were missing. We now included the citations in l. 48-49 of the main manuscript:

*“This is achieved by allowing multiple (max. 6, see **Supplementary Fig. 2**) imagers to bind simultaneously, maximizing their association rate (k_{on})¹⁷⁻²¹,...”*

Furthermore, we have also included the citation of Clowsley et al.⁹ reporting photo-induced docking strand loss in l. 83-84 of the main manuscript:

*“The 20 % decrease in TH detection over time is likely due to photo-induced damage to the DNA caused by reactive oxygen species^{21,23} during imaging (see **Supplementary Note 1**).”*

And in the Supplementary Note 1:

“The source of the damage lies in reactive oxygen species, confirming previous results^{1,3,10}”

2.4) Could even higher repeat numbers be beneficial for the tracking handle? A priori it not clear that 6 is “optimal” in any particular sense. Even though strands get longer the effective hydrated radius of the mostly single stranded handle grows slowly due to random coiling (as noted in Supplementary Figure 1) and higher repeat numbers could further increase robustness against photo-induced site loss. In connection with this point, more than one tracking handle could be employed per tracking particle. This would also allow lowering imager concentration further if desirable. Please discuss pros and cons.

We are very grateful for this valuable comment, which we think helped us to significantly strengthen the manuscript. As the reviewer correctly states, the number of six binding sites per TH has been defined as the initial condition for this project, which is not claimed to be optimal in any particular sense. The optimization process refers to examining the remaining parameters once the TH design has been fixed (e.g., buffer ion composition, temperature, imager concentration).

In line with our previous answers regarding that it is favorable to operate the TH at the lowest possible imager concentrations, particularly concerning cellular targets, we have now included new experiments examining the effect of using 2×TH per target molecule compared to 1×TH labeling and found that this allowed us to reduce imager concentrations by 8-fold without losing performance. We have added **Supplementary Fig. 23** and present the results in the main text, including a discussion of the cost of increased weight (see l. 153-162):

*“One approach to regain longer trajectory durations even at lower imager concentrations is to label each target molecule with multiple THs. We therefore designed DNA origami with two TH extensions (2×TH) and compared the effect on the recorded trajectory durations to standard TH origami (1×TH) for varying imager concentrations (**Supplementary Fig. 23**). While the weight imposed on the target molecule by the 2×TH labeling is only doubled, the observation times are dramatically increased, allowing an 8-fold reduction in imager concentration to measure similar observation times as for the 1×TH labeling. We reason that we can achieve a similar effect by extending the 1×TH handle sequence by multiples of the triplet CTC (e.g., from 18× to 36×). Naturally, both ways lead to an increased size and weight of the label, potentially interfering with the dynamics of the target molecules at a certain point or reducing the achievable localization precision. However, we think that this approach can be a viable starting point for further optimization, especially in cases where background fluorescence and/or unspecific binding are the limiting factors.”*

Supplementary Figure 23. 2xTH vs. 1xTH labeling. (a) Standard TH origami design with a single TH (1xTH) vs. origami featuring two TH labels at a 20 nm spacing (2xTH). (b) $T_{1/2}$ vs. imager concentration for immobilized 1xTH origami (standard, orange) and 2xTH origami (purple). Labeling with two THs dramatically increases observation times. The black dashed line highlights that using an imager concentration of 5 nM with 2xTH origami yields a similar $T_{1/2}$ compared to 1xTH origami at an imager concentration of 40 nM, hence, allowing an 8-fold reduction in imager concentration. (c) Complementary cumulative distribution functions (CCDF) of the occupancy for 1xTH origami (orange shades) and 2xTH (purple shades) for the same data sets as in (b). Error bars in b correspond to relative error determined in **Supplementary Fig. 6**.

2.5) In supplementary fig. 4 some of the description seemed difficult to follow. Particularly, the following sentences were hard to parse: “Second, all picks exceeding the 90% percentile of the number of events distribution of all picks are disregarded, i.e. all fluorescence traces consisting of more than 5 events in this case”; “For TH origami (illustration analogous to a) we divided the total number of localizations within a fluorescence trace by the measurement duration in frames (occupation) that can be visualized as a compression of the overall event durations”. Please consider reformulating. It seems odd to call this a compression.

We apologize for our unclear description in **Supplementary Fig. 4**. The figure and its caption have been rephrased accordingly, replacing the misleading term “compression”. This allowed us to already introduce the term occupancy, which was also brought up by Reviewer #1 and to which we refer to in the newly added **Supplementary Fig. 21 & 22**.

Supplementary Figure 4. Filter - immobilized. (a) The filtering procedure (see **Supplementary Fig. 3**) for immobilized SD origami is illustrated for an exemplary fluorescence trace (blue line). Valid picks (i.e. picks passing the filter criteria) yielded fluorescence traces with at least one localization within the first 5 frames (black line) of the measurement (blue area below trace). The bar below indicates the number of trajectories, i.e., continuous and uninterrupted fluorescence signal (here 3 trajectories). (b) Cumulative distribution function of number of trajectories per SD origami for an exemplary data set. As expected, the majority of SD origami yielded only a single trajectory before undergoing photobleaching. However, a small fraction of SD origami exhibited blinking behavior, causing interruptions in the fluorescence trace and hence an increased number of registered trajectories (as in the example in a). To remove potentially damaged/imperfect dye molecules, we discarded all picks exceeding the 90%-percentile (black line) of the distribution of number of trajectories of all picks, in this case picks exhibiting more than 5 trajectories. (c) For immobilized TH origami we calculated the ratio between the total time of a TH in the fluorescent state (i.e. the sum over all trajectory durations) and the total measurement time, which we define as the occupancy. In other words, the occupancy indicates the total time in which a TH is occupied with a fluorescing imager as a percentage of measurement time. Only picks with an occupancy of more than 20 % (black line) were used for further analysis. The black dashed line indicates a zoom into the fluorescence signal shown in (d). Analogous to (a), we additionally only considered TH origami picks yielding at least one localization within the first 5 frames (black line) of the measurement (red area below trace).

References

1. Chung, K. K. H. *et al.* Fluorogenic probe for fast 3D whole-cell DNA-PAINT. *bioRxiv* 2020.04.29.066886 (2020). doi:10.1101/2020.04.29.066886
2. Mücksch, J. *et al.* Quantifying Reversible Surface Binding via Surface-Integrated Fluorescence Correlation Spectroscopy. *Nano Lett.* **18**, 3185–3192 (2018).
3. Blumhardt, P. *et al.* Photo-Induced Depletion of Binding Sites in DNA-PAINT Microscopy. *Molecules* **23**, 3165 (2018).
4. Oi, C. *et al.* LIVE-PAINT allows super-resolution microscopy inside living cells using reversible peptide-protein interactions. *Commun. Biol.* **3**, 1–10 (2020).
5. Eklund, A. S., Ganji, M., Gavins, G., Seitz, O. & Jungmann, R. Peptide-PAINT Super-Resolution Imaging Using Transient Coiled Coil Interactions. *Nano Lett.* **20**, 6732–6737 (2020).
6. Sirghi, L., Kylián, O., Gilliland, D., Ceccone, G. & Rossi, F. Cleaning and Hydrophilization of Atomic Force Microscopy Silicon Probes. *J. Phys. Chem. B* **110**, 25975–25981 (2006).
7. Wiegand, G., Arribas-Layton, N., Hillebrandt, H., Sackmann, E. & Wagner, P. Electrical Properties of Supported Lipid Bilayer Membranes. *J. Phys. Chem. B* **106**, 4245–4254 (2002).
8. Civitci, F. *et al.* Fast and multiplexed superresolution imaging with DNA-PAINT-ERS. *Nat. Commun.* **11**, 4339 (2020).
9. Clowsley, A. H. *et al.* Repeat DNA-PAINT suppresses background and non-specific signals in optical nanoscopy. *Nat. Commun.* **12**, 501 (2021).

Reviewers' Comments:

Reviewer #1:

Remarks to the Author:

The authors have addressed my concerns, with a very compelling final manuscript.

Reviewer #2:

Remarks to the Author:

I appreciate the constructive approach that the authors have taken in addressing the comments that were raised. I am satisfied with the revised MS and am delighted to see that the additional handle approach seems to be a valuable option for optimisation.

REVIEWER COMMENTS

Reviewer #1 (Remarks to the Author):

The authors have addressed my concerns, with a very compelling final manuscript.

Reviewer #2 (Remarks to the Author):

I appreciate the constructive approach that the authors have taken in addressing the comments that were raised. I am satisfied with the revised MS and am delighted to see that the additional handle approach seems to be a valuable option for optimisation.

We would like to thank both reviewers for their positive feedback and again for their constructive and helpful comments throughout this revision process.